# Knowledge, Attitudes, and Perceived Barriers toward Genetic Testing and Pharmacogenomics among Healthcare Workers in the United Arab Emirates: A Cross-Sectional Study

**DOI:** 10.3390/jpm10040216

**Published:** 2020-11-09

**Authors:** Azhar T. Rahma, Mahanna Elsheik, Bassam R. Ali, Iffat Elbarazi, George P. Patrinos, Luai A. Ahmed, Fatma Al Maskari

**Affiliations:** 1Institute of Public Health, College of Medicine & Health Sciences, UAE University, Al Ain P.O. Box 17666, UAE; 201280026@uaeu.ac.ae (A.T.R.); mahanna.s@uaeu.ac.ae (M.E.); ielbarazi@uaeu.ac.ae (I.E.); luai.ahmed@uaeu.ac.ae (L.A.A.); 2Zayed Center for Health Sciences, UAE University, Al Ain P.O. Box 17666, UAE; bassam.ali@uaeu.ac.ae (B.R.A.); gpatrinos@upatras.gr (G.P.P.); 3Department of Pathology and Genomics and Genetics, College of Medicine and Health Sciences, UAE University, Al Ain P.O. Box 17666, UAE; 4Department of Pharmacy, School of Health Sciences, University of Patras, 26504 Patras, Greece

**Keywords:** genetics, medical, genetic counseling, health personnel attitudes, pharmacogenomics

## Abstract

In order to successfully translate the scientific models of genetic testing and pharmacogenomics into clinical practice, empowering healthcare workers with the right knowledge and functional understanding on the subject is essential. Limited research in the United Arab Emirates (UAE) have assessed healthcare worker stances towards genomics. This study aimed to assess healthcare workers’ knowledge and attitudes on genetic testing. A cross-sectional study was conducted among healthcare workers practicing in either public or private hospitals or clinics as pharmacists, nurses, physicians, managers, and allied health. Participants were recruited randomly and via snowball techniques. Surveys were collected between April and September 2019; out of 552 respondents, 63.4% were female, the mean age was 38 (±9.6) years old. The mean knowledge score was 5.2 (±2.3) out of nine, which shows a fair level of knowledge. The scores of respondents of pharmacy were 5.1 (±2.5), medicine 6.0 (±2.0), and nursing 4.8 (±2.1). All participants exhibited a fair knowledge level about genetic testing and pharmacogenomics. Of the respondents, 91.9% showed a positive attitude regarding availability of genetic testing. The top identified barrier to implementation was the cost of testing (62%), followed by lack of training or education and insurance coverage (57.8% and 57.2%, respectively). Building upon the positive attitudes and tackling the barriers and challenges will pave the road for full implementation of genetic testing and pharmacogenomics in the UAE. We recommend empowering healthcare workers by improving needed and tailored competencies related to their area of practice. We strongly urge the stakeholders to streamline and benchmark the workflow, algorithm, and guidelines to standardize the health and electronic system. Lastly, we advocate utilizing technology and electronic decision support as well as the translational report to back up healthcare workers in the UAE.

## 1. Introduction

There has been a considerable amount of research on genes and medications documenting variations in drug response in individuals. An individual’s genetic makeup greatly impacts their response to the medication, accounting for an estimated 20–95% of variations in drug response [1,2]. These findings give the premise to pharmacogenomics (PGX) and pharmacogenetics testing. The use of genetic tests to determine the ideal pharmaceutical therapy for a patient will improve drug efficacy and will reduce adverse drug responses [3,4,5]. The terms PGX and pharmacogenetics are often used interchangeably, but PGX has a larger focus on the entire genome’s influence on drug response [6].

According to the United Arab Emirates (UAE) Ministry of Health and Prevention (MOHAP), genetic disorders are ranked the fourth-highest cause of deaths in the UAE. Among 193 countries, the UAE is ranked sixth in the prevalence of birth defects, primarily due to genetic causes [7,8]. PGX and genetic testing can act as an important tool in understanding genetic makeup, recognizing disease-causing genes and providing preventive and supportive measures to these diseases.

The future of PGX implementation in medical practice is highly dependent on healthcare workers’ acceptance and the request of pharmacogenetics tests [5,9,10]. Pharmacists are suggested to be at the heart of PGX implementation due to their integral and unique roles as educators to healthcare workers and patients [4,6,11,12]. In fact, pharmacists demonstrated more positive perceptions than doctors or physicians toward PGX, as reported in two previous studies in Qatar and Kuwait [13,14]. In these studies, the majority of survey respondents were aware of the importance of PGX in individualized medicine. As the largest group in the healthcare workforce, nurses also play a central role in patient advocacy as defined by the American Nursing Association. Therefore, they are expected to be knowledgeable on this type of genetic testing to assume responsibility in integrating it in clinical practice [15].

The slow implementation of genetic testing and PGX can be attributed to many reasons, including but not limited to, a lack of evidence in clinical use, costing, and ethical concerns [4,15]. Despite the limited widespread implementation of PGX testing, it is currently being applied and used to guide treatments for certain cancers, diabetes, and cardiovascular diseases [16]. Cardiovascular disease (CVD) represents one of the foremost health threats in the UAE. According to the World Health Organization (WHO) report on the UAE, 40% of all deaths were due to cardiovascular diseases. The Department of Health in the emirate of Abu Dhabi (DOH) reported that 71% of the population has at least 1 CVD risk factor, predicting a rapid increase in future CVD events. Moreover, 12% of all deaths were due to cancers and 5% to diabetes [17,18].

Many barriers to PGX application have also been reported; however, lack of genomic knowledge and lack of healthcare professionals’ confidence in decision-making are widely prominent factors affecting the practice of PGX [4,5,19,20,21]. Therefore, it has been highlighted that more intense and advance PGX education and training is needed for healthcare professionals, especially pharmacists, for better the delivery of PGX and personalized medicine services [4,20,22,23].

Moreover, to successfully translate the discipline of PGX into clinical practice, all members of the healthcare workforce need to be knowledgeable and educated on the subject. To the best of our knowledge, there are no current research studies, to date, in the United Arab Emirates (UAE) assessing health professionals’ stance and attitudes towards PGX. We will use the Health Literacy Skills Framework (HLS) of Squiers et al. [24] to guide our assessment of the knowledge of healthcare workers. HLS is a comprehensive framework that appreciates the dynamic nature of knowledge and skills. The HLS had been conceptualized based on existing theoretical concepts and further developed by addressing the limitations of these concepts. The HLS framework primarily addresses the individual and ecological factors that influence knowledge, as well as the skills and stimulus that steer it [24]. This study aimed to assess the knowledge, attitudes, and perceptions of healthcare workers on PGX and personalized medicine, and their perceived barriers for full implementation of genetic testing and pharmacogenomics.

## 2. Materials and Methods

A cross-sectional study using a validated questionnaire was conducted. Inclusion criteria embodied registered healthcare workers practicing in either public or private hospitals or clinics. Registered pharmacists, nurses, physicians, managers, and allied health practitioners were invited as they were identified by literature as the stakeholders for the adoption of genetic testing and PGX. We accessed the online Shafafiya portal of the DOH that contains a population frame for all the healthcare providers working in the UAE (data included: Clinician License, Clinician Name, Major, Profession, Category, Gender, Facility Name, Facility License, Location, Facility Type, and the Status). Facilities were stratified per location and then contacted by the researcher either by email or by site visit to grant approval and distribution of the questionnaire among the healthcare providers. We employed random selection sampling and chain sampling techniques. The survey was offered both in person and via the internet. Due to the generally busy schedule of healthcare providers, some preferred answering the questionnaire on the spot while others preferred filling it out online at a later time when they were less busy. Moreover, some hospitals and clinics asked for the online survey so that they could circulate it to their healthcare providers via email, while other clinics asked for printed versions to be distributed by their human resources staff. Furthermore, the internet-based medium was used for snowball sampling. The survey was administered between April and September 2019 in order to reach the calculated target sample size. The survey was also kept open longer to accommodate the summer break period. This study was approved by the Social Science Research Ethics Committee of United Arab Emirates University (UAEU) ERS_2017_5671. Participants were asked to read the study’s information sheet and sign a consent form before answering the survey.

The questionnaire was designed based on previously validated and used tools to explore and identify knowledge, awareness, attitude, behavior, and interest in genetic testing and PGX [10,13,25].

We piloted the questionnaire among 50 medical and health sciences professionals and amended it accordingly. The questionnaire was administered in English and it was divided into 3 sections.

Section 1: Demographic data, e.g., age, gender, occupation, years of experience, and nationality. Section 2: Knowledge; we asked nine questions about specific genomic and PGX. A knowledge score was calculated from nine true and false questions about genetics and PGX. Three knowledge levels were created based on the number of correct answers: Good (7–9 correct answers), Fair (4–6 correct answers) and Poor (3 or less correct answers). Section 3: Attitudes of healthcare workers with regard to the ethical, social, and economic implications of genetic testing and PGX in addition to their perceived barriers for the full implementation of genetic testing and PGX in the UAE. For the attitudes, a 5-point Likert scale of strongly agree, agree, strongly disagree, disagree, and neutral was collapsed into agree, disagree, and neutral for ease of analysis and interpretation.

For statistical analysis, we calculated sample size using the formula for cross-sectional studies; (1.96^2^ × *P* (1 − *P*)/d^2^), where *P* = 0.27 (27% is the prevalence reported in similar previous studies) and d = 0.05. Sample size = 3.84 × 0.27 (1 − 0.27)/0.0025 = 303 healthcare workers. Accounting for an average response rate of 56–84% (reported in previous studies), the calculated sample size needed for this analysis was 444 healthcare workers.

International Business Machines Corporation Statistical Package for the Social Sciences (IBM SPSS) Statistics 26 was used for data analysis. Descriptive statistics (means, standard deviation, SD) and frequencies (percentages) were used to represent the data. Chi-squared test and Monte Carlo exact test were used to determine any significant differences in the distribution of respondents’ characteristics between the knowledge levels. Questions about genetics were selected from the literature with validated questions which recommended that the cutoff for good knowledge is 75%, and we followed the analysis of the literature in giving all the questions of the knowledge the same weights.

## 3. Results

Table 1 presents the respondents’ demographic characteristics. Out of 552 respondents, 63.4% were females. The mean age (±SD) was 38 (±9.6) years old, and 67.7% of the respondents were aged between 20 and 41 years old, and 26.9% between 20 and 30 years old. Most respondents had a pharmacy-related occupation (42%), followed by 52% belonging to either medicine or nursing occupations. More than half (52.2%) had over 10 years of experience.

### 3.1. Assessment of General Knowledge on Genetics and PGX

The mean knowledge score (SD) of the respondents was 5.2 (±2.3) out of nine, which shows a fair level of knowledge according to our scale. The mean knowledge score for respondents of pharmacy-related occupations was 5.1 (±2.5), medicine 6.0 (±2.0), and nursing 4.8 (±2.1). Respondents working in business and/or management positions and allied health professionals both had scores of 5.6 (±2.2 and ±1.1, respectively). Only 2 respondents out of 552 (0.4%) scored nine out of nine. For the second question, regarding nucleotide pairing, the percentage of respondents who answered correctly was only 1.7% higher than those who answered, “do not know.” A high percentage of 89.3% recognized correctly that genetic variances affect drug response. Table 2 summarizes the results of the general knowledge questions on genetics and PGX.

Table 3 summarizes the distribution of the levels of knowledge between different characteristics of the healthcare workers. The knowledge levels were significantly different between men and women (*p* = 0.01). Moreover, significant differences in knowledge levels were found between occupation groups (*p* = 0.00), completion status of a PGX training or education (*p* = 0.01), and having a patient who asked about taking a genetic test in the last two years (*p* = 0.02).

### 3.2. Attitudes towards the Applications of PGX

#### 3.2.1. Attitudes on Genetic Testing

We found that 74% of respondents would consider having a genetic test themselves performed at some point in their lives. The vast majority of respondents (91.9%) exhibited a positive attitude regarding availability of genetic testing. More than half (57.6%) reflected a positive response towards the accessibility of genetic tests (Appendix A).

#### 3.2.2. Concerns and Ethics

A common concern expressed by 74.4% of the recruited healthcare workers was that genetic test results would affect the quality of the patient’s medical care. Among the sample, 71.5% believed that PGX could be exploited and used as a means of discrimination (Appendix A).

#### 3.2.3. Desire to Participate

Statements questioning interest in genetic testing and PGX research was met with more overall positive responses, where 68.2% of respondents expressed a desire to participate in genetic research. Of the respondents, 83.7% indicated they would be interested in attending a course or educational seminar on PGX, and 43.4% would like to donate genetic material to a biobank.

#### 3.2.4. Current and Future Outlook on PGX

On the subject of legal frameworks, only 47.7% agreed that policies and procedures exist in the field of genetic tests in the UAE, with 44% taking a neutral stance. When questioned on the future of medicine, 87.4% of respondents believed medicine will become more personalized, and 85.3% agreed in thinking the government should invest more money in genetic testing development. Moreover, 87.2% think that more time should be allocated to teaching PGX during studies. The majority of respondents, 83.9%, agreed that the expenses of genetic tests should be covered by insurance companies.

#### 3.2.5. Barriers to Implementation

Out of the 474 respondents who answered the question on barriers to implementation, of PGX testing in the UAE, 62% identified the cost of testing as being a major barrier. Lack of training or education and insurance coverage followed as the second and third largest barriers (57.8% and 57.2%, respectively). Only 6.3% thought there was no clinical need for PGX testing.

#### 3.2.6. Type of Preferred Education

Out of 472 respondents, a majority (73.9%) chose workshops or seminars as their preferred learning method on PGX. Blended and internet-based learning received a similar reception to each other (30.9% and 27.3%, respectively).

### 3.3. Assessment of Personal Knowledge and Attitudes

When questioned on their own personal experience with genetic testing and PGX, 39.9% stated that PGX was a factor in their current work and 33.5% stated it was not. Less than half (41.7%) agreed when asked on whether they would be able to explain, without external elaboration, the results of genetic tests to their patients. Only 38.4% believed their undergraduate studies provided them with sufficient knowledge on genetics and PGX. Only 31% of respondents reported advising at least one of their patients to undertake a genetic test, as opposed to 43.2% of respondents reporting they have not previously advised it. The majority (64.5%) reported that patients have not asked about taking a genetic test in the last two years. Only 32.5% stated that patients asked for their advice on genetic test results in the last two years. When asked about who they thought should provide counseling on genetic and pharmacogenetics testing and results, 51.5% selected genetic counselor and 35.9% selected physician. Only 9.3% believed a pharmacist should assume this role.

## 4. Discussion

Assessing the knowledge and attitudes of the frontline workers of the health system is imperative for the seamless implementation of genetic testing and PGX. In the UAE, there are strides to implement genetic testing and pharmacogenomics; therefore, our findings will delineate the stringent approach of implementation. In our cohort, we assessed the knowledge and attitudes of the entire cluster of the healthcare workers including physicians, pharmacists, nurses, allied health and administrative workers as the stakeholders in the UAE foresee a multidisciplinary approach for the implementation of genetic testing and PGX. All participants in our cohort exhibited a fair knowledge level about genetic testing and pharmacogenomics. Most of the respondents showed a positive attitude regarding availability of genetic testing. The top identified barrier to implementation was the cost of testing followed by lack of training or education and insurance coverage.

Advances in genetic testing facilitated discovering genetic variants, which guided the drug prescription and tailored dose selection and replaced the trial and error approach. In fact, several guidelines and algorithms are incorporating and adopting pharmacogenomics in their clinical pathways, which in turn paved the road to personalized medicine [26,27,28,29,30,31,32]. Studies signify that physicians immersed in pharmacogenetics modules were more auspicious towards genetic testing as they sought it clinically beneficial. Furthermore, their awareness fueled their confidence in their skills to implement personalized medicine into their patient-centered care [33]. In her paper, Swan M. highlighted personalized medicine as one of the plans and routes for the Health Vision of 2050 [34]. In their paper, Mason-Suares, H. et al. [35] highlighted the new spectrum of skills required from healthcare providers in order to implement personalized medicine; some of these skills include managing diagnostics facilities, gauging the relevance of tests, and implementing cost-effective procedures. In our paper, we assessed the knowledge and attitude of healthcare workers in the UAE to gauge their position within the personalized medicine spectrum. We aimed to provide stakeholders in the UAE with the information needed to strategize their implementation approaches. From our findings, stakeholders should prioritize educating healthcare providers about the basics of genetics and translational aspects.

Studies have consistently demonstrated a gap in the knowledge of healthcare workers about genetic testing and PGX in almost all countries: United Kingdom, Greece, Canada, USA, Japan, Germany, Netherlands, Egypt, Africa, Brazil, Qatar, Kuwait, and KSA [10,14,33,36,37,38,39,40,41,42,43]. Similarly, our findings fall along the same line.

Interestingly, this study shows significant differences in the levels of theoretical knowledge of genomics and PGX by gender. The proportion of healthcare workers with good knowledge levels was higher in male than female workers, while more females scored moderate or fair knowledge levels than male healthcare workers. One study by Powell, K.P et al. [44] reported that the inconsistent levels of knowledge and understanding are significantly associated with gender. However, in their study, male workers were two times more likely to feel prepared to answer questions related to direct-to-consumer genetic tests than female workers. Gender gap of knowledge had been addressed in other scientific domains, but not in genetic testing and PGX. Many studies highlighted the reversed gender gap in education. This disparity warrants in-depth investigation and further research; as such, it requires a pivotal strategy [45,46].

Our study revealed significant statistical differences in the levels of genomics knowledge between different occupations. Respondents working in the field of medicine scored higher than those working in the field of pharmacy or nursing. However, all exhibited a fair knowledge level. In part, this can be attributed to the narrow application of genomics in the field of medicine in the UAE [47,48,49,50,51,52,53].

Remarkably, in our sample, knowledge scores for genomic basics was significantly associated with healthcare workers having patients asking them about undertaking a genetic test in the last two years. Notably, this was not the case if the patients asked them for advice about the results of a genetic test. This can potentially be explained by the fact that healthcare workers felt responsible and duty-bound to learn more about genomics and genetic tests to maintain the physician–patient rapport [54,55]. Another significant attribute to the knowledge of the healthcare workers is completing a training or education in genetic testing or PGX. A survey on Canadian physicians working in oncology, cardiology, and family medicine concluded that physicians with prior training on genomics medicine had a significantly higher mean knowledge score [56]. In fact, education and training are the foundation of most of the platforms, frameworks, and consortia that coined the implementation of genetic testing and personalized medicine [57,58,59,60].

Studies have repeatedly reported the positive attitude towards genetic testing and PGX that resides among healthcare workers. Our study is in line with this finding. The vast majority of respondents in our cohort exhibited a positive attitude regarding availability of genetic tests, biobank, and application of genetic testing and PGX. A review by Yau, A. et al. [40] concluded that doctors working in the USA, Canada, Japan, Germany, and the Netherlands had a positive attitude toward pharmacogenetics, despite the poor knowledge. Another systematic literature review disclosed that healthcare specialists saw merit in PGX [15]. Moreover, a study on pharmacists working in Quebec (Canada) voiced that pharmacists were very optimistic about the prospective role of PGX [42]. In our cohort of healthcare workers in the UAE, a genetic counselor was voted higher for assuming the role and responsibilities of counseling on PGX and genomic test and results, followed by physicians. Only 9.3% believed a pharmacist should assume this role, thereby conflicting with the previous findings of pharmacists having a significantly more positive attitude than doctors toward assuming the roles and responsibilities of PGX application and counseling [14]. Our findings fall along the same line as the findings on pharmacists and physicians in Greece, wherein they reported feeling incapable of clarifying the results of PGX tests to their customers or patients, and the authors tied that to the low level of undergraduate education in genetics and PGX [10].

Most healthcare workers in the UAE have considered having a genetic test performed at some point in their career in order to make better-informed decisions about their respective interventions and treatments. Therefore, we can extrapolate the positive attitude toward perceived clinical utility of genomic results. A mixed-method approach conducted by Stark, Z. et al. [61] on Australian health professionals echoed that genetics professionals perceived higher clinical utility towards rapid genomic testing in neonatal and pediatric intensive care than the intensivists themselves. More than half of the healthcare workers in the UAE reflected a positive attitude towards the accessibility of online direct-to-consumer genetic tests, whereas primary-care workers in Italy deemed the direct-to-customer genetic tests for chronic complex diseases to not be clinically useful [62]. A systematic review of the literature regarding the standpoint of health professionals concluded that health professionals specializing in genetics were most likely to express concerns toward direct-to-consumer tests due to their deep knowledge in comparison with other healthcare workers [63].

The top barrier for the implementation of genetic testing and PGX in the UAE identified by our respondents was the cost of testing, followed by lack of training or education and insurance coverage, lack of clinical guidelines, insufficient infrastructure, and lack of laws governing privacy and confidentiality. Implementing genetic testing and PGX in the UAE will first require addressing the aforementioned barriers on both individual and systematic levels. Physicians in the USA echo similar opinions as those of healthcare workers in our sample, whereby they rated costs of gene-based therapies and genetic testing as the most significant barrier [64,65]. A study by Najafzadeh, M. et al. [66] investigated the barriers to integrating personalized medicine into clinical practice using a best–worst scaling choice experiment, and labeled both education and guidelines as barriers to the implementation of genetic testing.

A variety of studies echoed the role of pharmacists in leading the implementation of pharmacogenomics within their work settings [67,68,69,70,71,72]. Given the United Arab Emirates’ endeavors to follow a multidisciplinary approach for project implementation, ensuring harmony, commitment, and unity, including a large variety of healthcare worker specialties in our cohort was very important [73,74,75,76,77,78,79]. In the focus group discussion that we conducted among pharmacists working in the UAE, they voiced their preference to have a multidisciplinary approach to implement pharmacogenomics [80].

Aggregating all healthcare workers into one pool is a limitation in our study; we recommend conducting studies focusing on each specialty to insure in-depth and tailored assessments of the gaps in knowledge, attitudes, and existing challenges. Moreover, we recommend conducting qualitative studies as that will lead to opening the door to a more comprehensive understanding of the attitudes of healthcare workers in the UAE.

## 5. Conclusions

Our study set the stage for the stakeholders occupied with implementing genetic testing and PGX in the UAE. Healthcare workers are the front-liners and the champions of the implementation strategies. Therefore, mapping their knowledge, attitudes, and concerns toward genetic testing and PGX will direct the framework for implementation. Crossing and bridging the chasm of knowledge will steer the implementation. We therefore recommend launching Continuing Medical Education (CME) accredited workshops presenting case studies and blended learning for healthcare providers. We urge collaboration between academia and healthcare to utilize experts in the field, seeing as most healthcare workers in the UAE have not studied pharmacogenomics at as part of their tertiary education. The positive attitude of healthcare workers will facilitate and guide the implementation strategies by identifying multidisciplinary champions. We urge the integration of genetic counselors in the implementation modules to bridge the current gap in knowledge and ability to counsel patients. We urge the stakeholders to implement laws to protect the privacy and confidentiality of genetic test results to avoid discrimination by insurance companies. We propose streamlining and benchmarking the workflow, algorithms, and guidelines. We advocate better utilization of technology and imputing the electronic decision support to back up healthcare workers in the UAE.

## Figures and Tables

**Table 1 jpm-10-00216-t001:** Demographic characteristics of healthcare workers (*N* = 552).

	Count (Percentage)
**Gender**	
Female	350 (63.4%)
Male	202 (36.6%)
**Age Group**	
20–30	148 (26.9%)
31–41	225 (40.8%)
42–52	124 (22.5%)
53–63	53 (9.6%)
64–74	1 (0.2%)
**Occupation**	
Pharmacy-Related	232 (42%)
Nurse	153 (27.7%)
Medicine	134 (24.3%)
Business & Management	14 (2.5%)
Administration	5 (0.9%)
Allied Health	5 (0.9%)
Governmental	5 (0.9%)
Intern	2 (0.4%)
**Years of Experience**	
<10 years	265 (52.2%)
>10 years	149 (29.3%)
**Nationality**	
Middle East	226 (40.9%)
Asia	179 (32.4%)
United Arab Emirates (UAE)	68 (12.3%)
Africa	34 (6.2%)
Europe & Australia	16 (2.9%)
North America	14 (2.5%)
Gulf Cooperation Council (GCC) countries	8 (1.4%)

**Table 2 jpm-10-00216-t002:** Questions assessing pharmacogenomics and genetics knowledge among healthcare workers (*N* = 552).

Choose the Correct Answer:	Correct Answer	True *n* (%)	False *n* (%)	Do Not Know *n* (%)
1. Humans have 48 chromosomes.	False	196 (38.8%)	281 (55.6%)	28 (5.5%)
2. Adenine (A) only pairs with cytosine (C) and Thymine (T) only pairs with Guanine (G).	False	148 (29.3%)	183 (36.2%)	174 (34.5%)
3. Pharmacogenomics seeks to individualize therapy based on the patient’s genetic profile.	True	407 (80.6%)	32 (6.3%)	66 (13.1%)
4. Genetic changes can cause adverse reactions.	True	395 (78.2%)	45 (8.9%)	65 (12.9%)
5. Pharmacogenomics testing is recommended by FDA for certain drugs.	True	335 (66.3%)	16 (3.2%)	154 (30.5%)
6. Genetic changes can affect the patient’s response to certain drugs.	True	451 (89.3%)	16 (3.2%)	38 (7.5%)
7. Genes can be activated or deactivated by other genes.	True	379 (75.0%)	38 (7.5%)	88 (17.4%)
8. Every cell of the body contains the whole genome.	False	338 (66.9%)	67 (13.3%)	100 (19.8%)
9. Environmental factors, such as cigarette smoke, can affect gene activity.	True	379 (75.0%)	52 (10.3%)	74 (14.7%)

**Table 3 jpm-10-00216-t003:** Comparison of the level of knowledge between different groups.

	Level of Knowledge			
	Good	Fair	Poor	*p*-Value
**Gender**				0.01 *
Female	95 (27.1%)	196 (56.0%)	59 (16.9%)	
Male	74 (36.6%)	87 (43.1%)	41 (20.3%)	
**Age Group**				0.12 **
20–30	46 (31.1%)	73 (49.3%)	29 (19.6%)	
31–41	63 (28.0%)	119 (52.9%)	43 (19.1%)	
42–52	34 (27.4%)	71 (57.3%)	19 (15.3%)	
53–63	25 (47.2%)	19 (35.8%)	9 (17.0%)	
64–74	1 (100%)	0 (0.0%)	0 (0.0%)	
**Years of Experience**				0.88
<10	72 (30.0%)	126 (52.5%)	42 (17.5%)	
>10	97 (31.1%)	157 (50.3%)	58 (18.6%)	
**Occupation Category**				0.00 **
Pharmacy-Related	69 (29.7%)	117 (50.4%)	46 (19.8%)	
Nurse	31 (20.3%)	88 (57.5%)	34 (22.2%)	
Medicine	61 (45.5%)	59 (44.0%)	14 (10.4%)	
Business & Management	5 (35.7%)	7 (50.0%)	2 (14.3%)	
Administration	0 (0.0%)	3 (60.0%)	2 (40.0%)	
Allied Health	1 (20.0%)	4 (80.0%)	0 (0.0%)	
Governmental	0 (0.0%)	3 (60.0%)	2 (40.0%)	
Intern	1 (50.0%)	1 (50.0%)	0 (0.0%)	
**Previous Exposure to Genetic Issues**				0.30
Yes	54 (35.3%)	75 (49.0%)	24 (15.7%)	
No	115 (28.8%)	208 (52.1%)	76 (19.0%)	
**Completed PGX/ Pharmacogenetics Training or Education**				0.01 *
Yes	51 (41.5%)	55 (44.7%)	17 (13.8%)	
No	118 (27.5%)	228 (53.1%)	83 (19.3%)	
**Have you ever advised any of your patients to undertake a genetic test?**				0.31
Yes	57 (38.0%)	83 (55.3%)	10 (6.7%)	
No	71 (34.6%)	112 (54.6%)	22 (10.7%)	
**Have you had any patients who asked about undertaking a genetic test in the last two years?**				0.02 *
Yes	59 (45.7%)	62 (48.1%)	8 (6.2%)	
No	74 (31.6%)	132 (56.4%)	28 (12.0%)	
**Have you had any patients who asked your advice about the results of a genetic test in the last two years?**				0.28
Yes	50 (41.7%)	57 (47.5%)	13 (10.8%)	
No	85 (34.1%)	140 (56.2%)	24 (9.6%)	

* significant value from Chi-square test. ** significant value from Monte Carlo exact test.

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
