# Peer review of "Knowledge, Attitudes, and Perceived Barriers toward Genetic Testing and Pharmacogenomics among Healthcare Workers in the United Arab Emirates: A Cross-Sectional Study"

_jpm, 2020, doi:10.3390/jpm10040216_

Round 1

Reviewer 1 Report

This is a cross-sectional survey of healthcare workers' knowledge, attitudes, and barriers to implementation of pharmacogenomics in United Arab Emirates. The survey was administered either in person or online. The following comments and recommendations are offered in order to strengthen the manuscript.

Title: None of the questions in the knowledge survey used the term "genomic medicine." Therefore, consider removing it.

Abstract: there is no mention of the target population of the survey. Please include the types of healthcare workers surveyed. At line 27, "All exhibiting a fair knowledge level." is not a complete sentence. At line 28, what was the positive attitude exhibited? At line 30, very colloquial language used. Consider developing a strategy for addressing and measuring barriers discovered.

Key words: very generic and non-specific. Remove implementation, barrier, knowledge, and attitude. Add MeSH terms 'genetics, medical,' 'genetic counseling,' 'health personnel attitudes,' and 'pharmacogenomics.'

In general, all previous studies lack a conceptual or theoretical framework for understanding and predicting knowledge and attitudes of healthcare workers. There are a number of frameworks that could be used for this purpose.

The figure at line 32 seems to be a pictorial of the authors' conclusion for a strategic approach with policy recommendations for implementing change to improve the uptake of genomic utilization in practice. It should be labeled as such without the caption above it, and placed in the discussion section.

Line 37-38: References 1 and 2 were cited to support the range of variable medication response based on genetic factors. However, the citations discuss the knowledge and experience of health professionals. References that support this epidemiology are suggested. Consider the following:

Chanfreau-Coffinier C, Hull LE, Lynch JA, et al. Projected Prevalence of Actionable Pharmacogenetic Variants and Level A Drugs Prescribed Among US Veterans Health Administration Pharmacy Users. JAMA Netw Open. 2019;2(6):e195345. doi:10.1001/jamanetworkopen.2019.5345  

Van Driest  SL, Shi  Y, Bowton  EA,  et al.  Clinically actionable genotypes among 10,000 patients with preemptive pharmacogenomic testing.  Clin Pharmacol Ther. 2014;95(4):423-431. doi:10.1038/clpt.2013.229

Bush  WS, Crosslin  DR, Owusu-Obeng  A,  et al.  Genetic variation among 82 pharmacogenes: the PGRNseq data from the eMERGE network.  Clin Pharmacol Ther. 2016;100(2):160-169. doi:10.1002/cpt.350  

Methods: How did you identify the population from which the random sample was generated? Why were two different surveying methods, in-person and online, needed? Why were all the questions about genetic facts given the same weight in the analysis? Did the pharmacy-related category contain pharmacists, technicians, and assistants? If so, why were different levels of education within the same worker category lumped together? What specialty(ies) of physicians were surveyed? Were all nurses registered, practical, or aides? Why did you need to have the survey open for 6 months?

Line 96: what are knowledge levels?

Results: how did you sort out an attitude from a belief? In Figure S1, you use the label, 'views and considerations.' This seems to imply opinions rather than attitudes. What did you measure?

Conclusion: quite colloquial and jargonish in style. Must be reworded and follow from the results.

Refernces: mdpi style is used.

Reviewer 2 Report

The authors have conducted a cross-sectional study for the investigation of Knowledge, Attitude, and Perceived Barriers toward 2 Genomic Medicine and Pharmacogenomics among 3 Healthcare Workers in the United Arab Emirates. They have attempted to find patterns of attitudes towards new and cutting-edge disciplines among healthcare professionals.

Major comments:

First of all the authors should declare their rationale behind their study. Why did they consider it to be of significance to study this topic in the local scientific population. This could be added as a paragraph in the "Introduction" section.

It is also very important to add two more paragraphs in the "Discussion" section:

first, they should discuss on the issue of how awareness on PGX and its advantages is expected to benefit personalized medicine, as well as why is important for healthcare professionals to be aware of cutting-edge technologies.

Second, they should address the different gravity of healthcare professional specialties on PGX awareness. For example, which is more important and whose opinion is more definitive, those of pharmacists, nurses etc. physicians or basic scientists.

Minor comments:

line 59: please replace the "...science of PGX..." with "...discipline of PGX..." since pharmacogenomics is not a science but a scientific field/discipline.

Some minot English corrections are required.

Round 2

Reviewer 1 Report

I appreciate the opportunity to review this much improved manuscript. I hope that it will have the impact and importance that the authors suggest. PGx is part of an evolving international health work up for individualizing drug therapy. Congratulations for a meaningful contribution to the literature.

Reviewer 2 Report

The authors have adressed all my prvious comments.